# Of Orchids and Dandelions: Empathy but Not Sensory Processing Sensitivity Is Associated with Tactile Discrimination Abilities

**DOI:** 10.3390/brainsci12050641

**Published:** 2022-05-12

**Authors:** Michael Schaefer, Marie-Christin Kevekordes, Hanna Sommer, Matti Gärtner

**Affiliations:** Fakultät Naturwissenschaften, Medical School Berlin, 14197 Berlin, Germany; marie.kevekordes@medicalschool-berlin.de (M.-C.K.); hanna.sommer@medicalschool-berlin.de (H.S.); matti.gaertner@medicalschool-berlin.de (M.G.)

**Keywords:** empathy, social, tactile, sensory processing sensitivity, personality

## Abstract

Many concepts of the human personality are based on assumptions about underlying physiological processes. The most prominent example is probably the concept of extraversion introduced by H.J. Eysenck decades ago. However, more recent approaches also propose that personality traits may be reflected by physiological processes. For example, empathic personality dimensions have been linked to tactile perception, suggesting that individuals with higher tactile sensitivity are also more empathetic to the sensations of others. Another recent example is the concept of sensory processing sensitivity, which has been linked to enhanced primary sensory processing. However, the exact relationship between tactile abilities and personality is still unclear, thus the current study aims to test whether different personality dimensions affect the performance in a tactile acuity task. Tactile abilities of healthy participants were tested with tactile 2-point-thresholds on the hands. Personality dimensions were examined with respect to empathy, sensory processing sensitivity, and the Big Five. Results revealed that empathy, but not sensory processing sensitivity, was associated with tactile performance. We conclude that the ability to feel with someone else seems to be linked to the perception of our own body. Thus, the sense of touch may play an important role for empathy. We discuss explanations of these results and highlight possible implications of our findings.

## 1. Introduction

The sense of touch is probably one of the earliest sensory systems to emerge in evolution. The tactile system occurred many million years before eyes and ears were developed in more complex multicellular animals [1,2]. Through the sense of touch, simple invertebrates not only receive information about the physical condition of the immediate environment, but also the possibility of the first basic forms of social action. Thus, touch is perhaps the most elementary form of communication. Humans may have much more sophisticated tools for social behavior (e.g., language), but still, the tactile sense is important for us when we interact in the social world. For example, we touch someone to say hello, to start a romantic relationship, or when we try to soothe him or her (e.g., [3,4,5]).

Considering that we behave differently in the social world, it seems reasonable that the relationship between our (tactile) senses and perception and behavior in social life varies according to personality traits. One of the earliest theories argued that differences between introversion and extroversion might be explained by variability in cortical arousal [6]. In Eysenck’s theory introverted individuals are permanently more aroused than extroverted people, resulting in an inherent drive to compensate for the high cortical arousal. This arousal is thought to be produced in the reticular formation and visceral brain. Based on this theory, introverts should have lower sensory thresholds in the ascending reticular activation systems (ARAS). Eysenck hypothesized that a high cortical arousal facilitates the detection of weak stimuli. Psychophysiological studies supported this assumption, showing, for example, lower auditory and tactile thresholds associated with introversion [7].

Another more recent approach hypothesized that empathic personality dimensions might be related to our tactile sense, e.g., the somatosensory cortices. At first glance, this seems surprising, given that primary (SI) and secondary (SII) somatosensory cortices in the postcentral gyrus are well-known to represent touch on the own body [2]. However, research in the last decades have consistently demonstrated that even observed touch on other’s bodies engages our own somatosensory cortices, suggesting that we feel and understand seen touch through simulation processes [8,9,10,11,12,13,14]. Moreover, it has been shown that dispositional empathy is associated with performance in a tactile acuity task, thereby suggesting that we may emotionally understand the other better when we are good at processing tactile stimuli felt on our own body [15]. Furthermore, research in mirror touch synesthesia similarly shows that in some individuals an observed tactile sensation can produce a felt tactile sensation on their own body, suggesting that we empathize with others through a process of simulation [16,17].

Furthermore, recent theories argue that there may be general individual differences in environmental sensitivity, suggesting different underlying brain processes when processing ecological stimuli [18]. One example is the theory of sensory processing sensitivity (SPS) [19]. This approach argues that there are interindividual differences with respect to the sensitivity both to aversive and supportive environmental stimuli, which have led some researchers to call extremes of these personalities as *orchids and dandelions* (e.g., [20]). Lionetti et al. used these flower metaphors to characterize highly sensitive individuals as orchids and less sensitive people as dandelions [20]. According to Aron et al., SPS can be described as a personality trait (rather than a disorder), which represents a significant survival strategy both for humans and animals [21,22]. It is assumed that individuals with high SPS have greater sensitivity to external and social stimuli, display higher emotional reactivity, and show greater depth of processing [19,22,23]. This has been supported, for example, by Acevedo et al., demonstrating higher brain activations for individuals with high SPS in brain regions known to be related to awareness and empathy, as well as higher-visual processing [24,25], suggesting that SPS represents an increased depth of processing [26]. It remains unclear whether SPS also affects the tactile modality, a fundamental sense for social perceptions. Furthermore, the proposed concept of SPS is still controversial discussed, in particular with respect to consider it as a personality trait [27].

Taken together, more research is needed to determine the relationship of empathic abilities and SPS to processing in the tactile modality. The present study aims to fill this gap by testing whether personality traits affect the performance in a tactile acuity task. Forty healthy participants were asked to complete questionnaires on the Big Five, SPS, and dispositional empathy. Then we tested tactile thresholds by employing a 2-point-discrimination task on the right and left hand. Based on previous research [14,15,17] and theoretical concepts [22] we hypothesized that empathy and SPS are associated with tactile acuity.

## 2. Materials and Methods

### 2.1. Participants

The study included 40 right-handed native German healthy volunteers (26 females) with a mean age of 29.50 years (±10.62 standard deviation). It adhered to the Declaration of Helsinki and was approved by the ethics committee of the Medical School Berlin (Germany). All participants gave written informed consent to the study and had no neurological or psychiatric history.

### 2.2. Materials and Procedure

To determine tactile thresholds of the participants, we used a commercially available discriminator (AFH-Webshop, Lügde, Germany; https://premium-therapie.de/de/befund-diagnostik/sensorik-senibilitaetstest/afh-2-punkt-diskriminator-duo, (accessed on 16 March 2022)). The discriminator consisted of 4 pairs of brass needles mounted on a rotatable disc that allowed switching rapidly between pairs. A single needle was used as the control condition. The space between the other needles ranged from 7 to 13 mm. Needles were applied on the palm of the left and right hand. The procedure was similar to previous studies [28,29]. At the beginning, we asked the subjects to close their eyes. Then we presented the stimuli ten times in a randomized order, without telling the participants about the ratio of needle pairs and single needles. Immediately after each trial the participant had to decide whether they have felt one or two sensations on the hand. In total, the session consisted of 160 trials for both hands. Finally, we calculated the number of correct responses for all trials to compute a score of tactile performance acuity.

On a separate day, we asked the participants to complete several personality questionnaires. Big Five personality dimensions were examined by using the Big Five Inventory 2 (BFI-2). The five-factor model of personality describes the personality traits extraversion, neuroticism, openness, conscientiousness, and agreeableness. The BFI is an established questionnaire to measure the Big Five, which shows good psychometric characteristics [30]. It consists out of 60 items, each has to be rated on a 5-point scale ranging from “do not agree at all” to “completely agree”.

Dispositional empathy was determined by employing a German version of the interpersonal reactivity index (IRI) that measures self-reported empathic behavior [31,32]. This 28-item questionnaire is well-known to measure trait empathy and is extensively validated (e.g., [33,34]). The IRI consists of four subscales with each pointing to different aspects of empathy. The scale empathic concern (EC) describes feelings of sympathy and concern for others, perspective taking (PT) assesses the tendency to cognitively imagine a situation from the other person’s point of view, imagination or fantasy (FS) is related to the ability to transpose oneself into the feelings and actions of fictional characters in books or movies, and personal distress (PD) taps the tendency to experience aversive feelings in response to distress in others. EC and PD describe an affective component, while FS and PT focus on a cognitive dimension of empathy [31].

SPS was measured using the highly sensitive person scale for German speaking populations (HSPS-G) scale [35], which is the German version of the HSPS, which has been developed by Aron et al. [19]. This questionnaire consists of 39 items and has been validated before [35].

### 2.3. Statistical Analyses

To test the relationship between personality and performance in the tactile acuity task, personality dimensions (SPS and empathy) were included as predictors in a multiple linear regression analysis with tactile performance as the dependent variable. We calculated a separate analysis for the Big Five personality dimensions. Furthermore, we computed separate analyses for left and right hand because our previous study found different results for left and right body side [15]. Since previous research has shown that age affects tactile performance, *age* was included as an additional predictor [36,37,38]. The software package SPSS was used for all statistical analyses (Version 27.1, IBM Corp., Armonk, NY, USA).

## 3. Results

Table 1 depicts the mean scores for IRI, BFI, and SPS. The performance in the tactile task was 88.31% ± 6.53% correct responses for the right hand and 88.72% ± 5.95% for the left hand. Left and right tactile acuity correlated with r = 0.68 (similar to our previous study [15]). Table 2 depicts Pearsons’ correlations of tactile performance with personality dimensions. Results demonstrate positive correlations of tactile acuity with empathy (empathic concern), but not with SPS or other personality dimensions (Bonferroni corrected, adjusted *p*-value of 0.005). Scatterplots are shown in Figure 1.

To further investigate the contribution of empathy and SPS on performance in the tactile acuity task, we calculated a linear regression analysis, in which all four empathy measures, SPS, and age went simultaneously into a model to predict tactile acuity of the right hand. Results showed a significant model (R = 0.70, adj. R^2^ = 0.39, F(6,39) = 5.19, *p <* 0.001) and demonstrated that empathic concern, perspective taking, and personal distress were predictors for tactile performance of the right hand (empathic concern: beta = 0.38, *p =* 0.026, perspective taking: beta = 0.30, *p =* 0.057; personal distress: beta = 0.36, *p <* 0.016). Furthermore, age was a significant predictor. In contrast, SPS was not a significant predictor for tactile performance (see Table 3).

To examine the robustness of this relationship between tactile acuity and empathy we calculated a sensitivity post-hoc analysis using G-power [39]. Results revealed a power (1 − beta error probability) of 0.98 (alpha = 0.05). 

The analogue calculation for left hand tactile performance showed weaker results. A linear regression model with predictors described above showed a significant model (R = 0.56, adj. R^2^ = 0.19, F(6,39) = 2.48, *p =* 0.04), but only with perspective taking as a predictor with significance at borderline (beta = 0.36, *p =* 0.048), see Table 4.

Big Five personality showed positive relationships with tactile thresholds of the right hand (neuroticism: r = 0.32, openness: r = 0.35), but failed to reach the level of significance. When calculating a regression model for tactile thresholds on the right hand with the Big Five and age as predictors, we found a significant model (R = 0.58, adj. R^2^ = 0.21, F(6,39) = 2.75, *p =* 0.03) but did not find any significant predictors of tactile thresholds other than age (extraversion: beta = −0.17, *p =* 0.31). For left hand thresholds we did not find significant correlations or regression model.

To further examine the personality trait SPS, we calculated a linear regression analysis including empathy (total score of IRI), age, neuroticism, openness, and extraversion as predictors. Results revealed a significant model (R = 0.77, adj. R^2^ = 0.60, F(5,39) = 10.10, *p <* 0.001) with neuroticism (beta = 0.43, *p =* 0.007) and empathy (beta = 0.34, *p =* 0.03) as strong predictors of SPS. All other dimensions failed to show significant effects (see Table 5). For all linear regression analyses multicollinearity (VIF scores) was low.

## 4. Discussion

Based on previous results and theoretical assumptions, this study aimed to examine the relationships of personality traits with the performance in a tactile task. Results demonstrated that tactile acuity was predicted by empathy personality traits, but not by SPS.

The results confirm previous findings about an association of tactile acuity and empathy personality traits. Similar to the present results, our previous study reported an association of empathy with tactile acuity of the right hand [15]. However, the former study examined tactile thresholds on the digits, whereas the present study applied touch to the palm of the hands. Thus, the current results extend previous findings by showing that tactile thresholds of other body parts also seem to be linked to empathic personality traits (at the right side of the body).

Why is the ability to feel with others linked to the tactile sensitivity of my own body? We argue that empathy can be understood as a simulation process. According to the perception-action model (PAM) we empathize with others by simulating their actions, sensations, or pain [40]. In this view, we understand touch seen on others’ bodies by the vicarious activation of our own somatosensory brain areas [41]. We further argue that the more attentive we are to our own bodily sensations, the better we are in simulating the touch seen on other bodies. Therefore, highly empathic individuals might also show lower tactile thresholds. However, these thoughts remain speculative, since we did not examine vicarious touch sensations but only found self-reported empathy associated with tactile sensory thresholds.

Nevertheless, previous studies also suggest that empathy might be related to tactile performance. For example, Philip et al. suggested that mental states may alter tactile performance. They investigated Zen scholars when exercising (meditation) for three days (focusing sustained attention on a body part) and found improved tactile thresholds for the body part in focus [28]. Meditation can be described as a method to train empathy [42]. Moreover, Banissy et al. demonstrated higher tactile acuity in mirror touch synesthesia, suggesting a hyper-sensitive perceptual system for those individuals. Thus, the authors argue that we emphasize with others through a process of simulation, suggesting that the tactile modality plays an important role for empathy [16,17]. In addition, it has been shown that dispositional empathy predicted activity in primary somatosensory cortex when receiving touch by a hand [43]. Furthermore, structural brain differences in somatosensory brain areas have been linked to dispositional empathy [44]. Moreover, several recent studies using imaging methods demonstrate a relationship between empathy and social touch [45,46,47,48], which is also supported by qualitative approaches, for example, in physicians (e.g., [49]). Thus, empathy and touch seem to be closely associated.

In contrast to empathy, SPS showed no correlation with tactile performance. It has been suggested that individuals with high SPS have greater sensitivity to external or social stimuli and may represent an increased depth of processing [22]. Previous research found support for this view with respect to visual processing and awareness [24,25]. However, the present results do not show a link between an enhanced tactile sensitivity and SPS. Future studies are needed to further examine whether improved sensory processing may only affect visual (or other modalities), but not the sense of touch. It may also be possible that SPS is related to other kinds of touch [50].

The concept of SPS is still controversial. Our results demonstrate that SPS is strongly linked to neuroticism and empathy, thereby confirming earlier studies [22]. However, we did not find a relationship of SPS with extroversion when using empathy and other personality traits as predictors. Further research is needed to determine whether SPS represents a unique trait or rather reflects a combination of some other personality dimensions. Given that SPS seems to be a rather heterogeneous concept addressing several distinct dimensions, future studies should consider other and newer questionnaires to measure (hyper)sensitivity more directly, for example, the Glasgow Sensory Questionnaire [51] or the Sensory Perception Quotient [52,53,54].

We further did not find any significant relationships of tactile sensory thresholds and Big Five personality dimensions. According to H.J. Eysenck, one could hypothesize that extraversion might be linked to enhanced tactile thresholds. Some older studies found support for this hypothesis. Edman et al. showed that tactile detection thresholds in introverts (and subjects with high neuroticism) are lower [7]. Furthermore, it has been reported that somatosensory evoked potentials are linked with the introversion dimension [55]. Although the present results found a negative relationship of extraversion with tactile thresholds, as predicted, the link failed to reach the level of significance. This may be explained by the limited sample size our study (e.g., Shagass and Schwartz included 89 subjects) and by differences of the samples (e.g., with respect of the range of age: Shagass and Schwartz found an interaction of extraversion and age with somatosensory processing). However, we found lower tactile thresholds for participants with high neuroticism scores, which is in line with Edman et al. [7]. Furthermore, our results are also in line with more recent studies suggesting relationships between early somatosensory processing and personality or dispositions to develop diseases [56,57,58].

Similar to our previous study we found relationships of tactile thresholds with empathy only for the right hand [15]. What is the reason for this laterality? Although it is well-known that the right side of our brain is linked to social perceptions, this does not mean that the left hemisphere is not important for processing social information. For example, it has been suggested that the left side of our brain is linked to emotions related to engaging and approaching, e.g., happiness when viewing a smile, whereas the right hemisphere may be associated with emotions related to avoidance, for example, fear [59]. Other reasons may point to handedness (all of our participants were right-handed). Future studies are needed to address this finding.

Several limitations of this study have to be taken into account. First, our sample size is rather small considering that we aim to examine personality dimensions. Second, only correlational data is reported. Third, based on the present data we are unable to give information about the neural underpinnings of the relationships we report. Fourth, empathy was measured by self-reporting questionnaires. Future studies should also consider actual empathic behavior to measure state empathy. However, we think that it is unlikely that compliance to the task may explain our results, since we did not find any correlations with the conscientiousness personality trait. Last, our study included many correlations, which increase the risk of type I error. We addressed this problem by adjusting probability values using Bonferroni correction.

We conclude that empathic personality traits seem to be associated with tactile sensory thresholds. We believe that these results could have important practical and clinical implications. One could speculate that individuals showing a lack of empathy might benefit from an unusual training to enhance the sensibility to their own body. For example, one might consider incorporating sensorimotor exercises into empathy training for psychopaths, which can help increase awareness of one’s own and others’ bodies [60]. Thus, the sense of touch, which is so important in our social lives, could be used to better understand how others feel.

## Figures and Tables

**Figure 1 brainsci-12-00641-f001:**
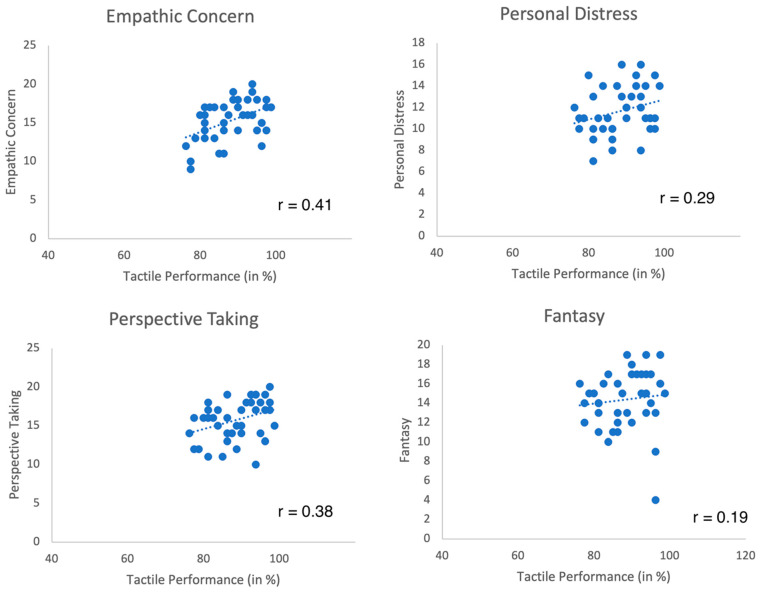
Scatterplots of tactile performance and empathy subscales (Pearson correlations). Empathy was linearly associated with tactile acuity (correlation coefficients of Fantasy raised to r = 0.37 when removing two outliers). SPS was not related to performance in the tactile task (see Table 2 for details).

**Table 1 brainsci-12-00641-t001:** Results of personality questionnaires SPS, IRI, and BFI.

		Mean ± Standard Deviation
**SPS**		90.93 ± 14.63
**BFI**	*Neuroticism*	13.13 ± 3.22
	*Extraversion*	15.05 ± 3.14
	*Openness*	21.38 ± 2.99
	*Agreeableness*	12.85 ± 3.42
	*Conscientiousness*	18.85 ± 2.87
**Empathy Personality Questionnaire IRI**	*Empathic Concern*	15.33 ± 2.59
	*Personal Distress*	11.65 ± 2.26
	*Perspective Taking*	15.68 ± 2.58
	*Fantasy*	14.35 ± 3.00

**Table 2 brainsci-12-00641-t002:** Correlation matrix of personality questionnaires and tactile performance (Pearson, significant results in bold).

	EC	FS	PT	PD	SPS	N	E	O	C	A	tact. r.
**EC**											
**FS**	0.50										
**PT**	0.45	0.37									
**PD**	0.20	0.34	−0.07								
**SPS**	**0.53**	**0.59**	0.39	0.31							
**Neuroticism**	**0.57**	0.47	0.34	0.35	**0.68**						
**Extraversion**	−0.11	0.15	−0.08	−0.22	−0.28	−0.35					
**Openness**	0.49	0.33	0.46	0.14	0.45	0.32	−0.04				
**Conscientiousness**	0.04	−0.07	0.23	0.02	−0.17	−0.13	−0.05	0.06			
**Agreableness**	0.10	0.02	0.11	−0.22	0.07	−0.09	0.16	0.04	−0.11		
**tactile acuity right hand**	**0.46**	0.11	0.35	0.27	0.14	0.32	0.07	0.35	−0.04	0.15	
**tactile acuity left hand**	0.27	0.25	0.34	0.26	0.08	0.26	0.01	0.14	−0.09	0.10	**0.68**

**Table 3 brainsci-12-00641-t003:** Regression analyses of tactile performance of the right hand with personality measures and age as predictors (significant results in bold).

Model					Coefficients (Standardized)		
R	R^2^	adj. R^2^	ANOVA		Betas	T	sign.
0.70	0.49	0.39	F(6,39) = 5.19, *p <* 0.001	EC FS PT PD SPS age	0.38 −0.29 0.30 0.36 −0.17 −0.36	2.34 −1.76 1.98 2.54 −0.98 −2.68	***p =* 0.02** *p =* 0.08 *p =* 0.06 ***p =* 0.02** *p =* 0.33 ***p =* 0.01**

**Table 4 brainsci-12-00641-t004:** Regression analyses of tactile performance of the left hand with personality measures age as predictors.

Model					Coefficients (Standardized)		
R	R^2^	adj. R^2^	ANOVA		Betas	T	sign.
0.56	0.31	0.19	F(6,39) = 2.48, *p =* 0.04	EC FS PT PD SPS age	0.07 0.09 0.36 0.32 −0.28 −0.25	0.38 0.45 2.05 2.00 −1.44 −1.63	*p =* 0.70 *p =* 0.66 *p =* 0.05 *p =* 0.05 *p =* 0.16 *p =* 0.11

**Table 5 brainsci-12-00641-t005:** Regression analyses of SPS with personality measures as predictors (significant results in bold).

Model					Coefficients (Standardized)		
R	R^2^	adj. R^2^	ANOVA		Betas	T	sign.
0.77	0.60	0.54	F(5,39) = 10.10, *p* < 0.001	Neuroticism Extroversion Openness Empathy (IRI) Age	0.43 −0.10 0.16 0.34 0.14	2.85 −0.77 1.24 2.21 1.12	***p* < 0.01** *p* = 0.45 *p* = 0.22 ***p* = 0.03** *p* = 0.27

## Data Availability

The data presented in this study are available on request from the corresponding author.

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
