# Peer review of "Of Orchids and Dandelions: Empathy but Not Sensory Processing Sensitivity Is Associated with Tactile Discrimination Abilities"

_brainsci, 2022, doi:10.3390/brainsci12050641_

Round 1

Reviewer 1 Report

This is a well-designed study that partially replicates and extends (using different personality-related measures) prior research.  It is an under-researched area that warrants further investigation.  My comments are all relatively minor.

“by Eysenck decades ago” – you should probably make clear it was HJ Eysenck given that there are two famous personality researchers with that name

“performance in a tactile acuity task seems to predict empathy” – but, of course, you could phrase it exactly the other way around.  I suggest a more neutral phrase around an association (e.g. a common latent construct).

The SPS is a very heterogeneous measure when you look at the items on it (e.g. some of them relate to introversion which is not directly a sensory phenomenon) and the researchers should contrast this with more modern measures where the focus is indeed in the sensory domain – e.g. Sensory Perception Quotient, Glasgow Sensory Questionnaire, Sensory Gating Inventory

I would like to see a post-hoc power analysis (aka sensitivity analysis) based on the observed effect sizes.

Author Response

Comments of reviewer #1:

We would like to thank the reviewer for the positive and helpful comments on our manuscript.

  1. „ “by Eysenck decades ago” – you should probably make clear it was HJ Eysenck given that there are two famous personality researchers with that name”

Thank you, in the revised version we corrected this point.

2.„ “performance in a tactile acuity task seems to predict empathy” – but, of course, you could phrase it exactly the other way around.  I suggest a more neutral phrase around an association (e.g. a common latent construct).

We agree with the reviewer and changed this point to a more neutral statement here:

“We conclude that the ability to feel with someone else seems to be linked to the perception of our own body.”

  1. „The SPS is a very heterogeneous measure when you look at the items on it (e.g. some of them relate to introversion which is not directly a sensory phenomenon) and the researchers should contrast this with more modern measures where the focus is indeed in the sensory domain – e.g. Sensory Perception Quotient, Glasgow Sensory Questionnaire, Sensory Gating Inventory”

Thank you for drawing our attention to this point. We fully agree with the reviewer and bring up this issue in the discussion section. We added on page 8:

“Given that SPS seems to be a rather heterogeneous concept addressing several distinct dimensions, future studies should consider other and newer questionnaires to measure (hyper)sensitivity more directly, for example, the Glasgow Sensory Questionnaire (Robertson & Simmons, 2013) or the Sensory Perception Quotient (Tavassoli, Hoekstra, & Baron-Cohen, 2014; Taylor, Holt, Tavassoli, Ashwin, & Baron-Cohen, 2020; van Leeuwen, van Petersen, Burghoorn, Dingemanse, & van Lier, 2019). ”

  1. „I would like to see a post-hoc power analysis (aka sensitivity analysis) based on the observed effect sizes.”

The revised version now includes a post-hoc power analysis, as suggested (page 5):

“To examine the robustness of this relationship between tactile acuity and empathy we calculated a sensitivity post-hoc analysis using G-power (Faul, Erdfelder, Lang, & Buchner, 2007). Results revealed a power (1 – beta error probability) of 0.98 (alpha = 0.05).”     

Reviewer 2 Report

“Of Orchids and Dandelions…” investigated correlations between a set of personality traits with tactile acuity. The manuscript is an interesting study that is well researched and well written. The study looked at personality traits centered on empathy to see how they correlate with tactile discrimination abilities. It evaluates the big five, plus Empathic concern, personal distress, perspective talking, fantasy, and sensory processing sensitivity. They found that tactile discrimination correlates with empathy but not sensory processing sensitivity.

Major comments:

The paper tests a lot of variables and should account for these multiple comparisons. In addition it separates tactile acuity by hand. The logic as to why this should matter and why they don’t include a measure of both hands is not clear and should by explained.. In addition they should provide the correlations for tactile acuity for both hands.

My other major concern is that the extant research is not discussed in a clear way. The paper includes SEP work from 1965, but EEG has evolved quite a bit since then. They should include similar studies from the last 10 years and build their arguments based on these.

Minor comments:

P1, L17 Remove (IRI)

 P1, L17 Remove (HSPS)

P1, L27  “to emerge” – remove -d

P1, L17 “that developed a type of tactile sense”, the tactile sense consists of several receptor types e.g. Meissner, Ruffini, Pacinian etc.

P1, L35 interact instead of act

P2, L48 “hypothesized”

P2, L50 The somatosensory system is more than just the cortex, please rephrase

P2, L51 in the postcentral gyrus (not on)

P2, L56 we are god at (not in)

P2, L57 You list studies (11-13) which seem very relevant. These should be described in detail

P2, L63 Since you decided to call the paper Orchids and Dandelions and include it here, maybe you should explain what it is

P2, L72

P2, L54 Taken together, more research is needed…

P2, L55 “or SPS to processing” (not with)

P3, L110 Spell out acronyms SPF and IRI

P3, L110 In abstract you say you used IRI not SPF

P3, L116 Fantasy is german, Imagination is a better word in English, methinks

P3, L118 Remove “According to Davis”

P3, L122 spell out HSPS

P3, L126 Why not FS, PT, PD?

P4, Do you need correction for multiple comparison?

P4, L151 Why not combine left and right hands?

P6 Discuss the logic of logic at handedness separately and why right hand

P6, Discuss how the many tests can influence results

P7, 214 stained? did you mean sustained?

P7, 215 the body part in focus

P7, 240 some of this research is quite ancient (Shagass and Schwarz, 1965). SEPs looked quite differ back then. Our modern systems are much better than 1965. Include more modern work.

Author Response

Comments of reviewer #2:

Thank you for the positive and helpful comments on our manuscript.

  1. „ The paper tests a lot of variables and should account for these multiple comparisons. In addition it separates tactile acuity by hand. The logic as to why this should matter and why they don’t include a measure of both hands is not clear and should by explained.. In addition they should provide the correlations for tactile acuity for both hands.”

 This is a good point. We agree with the reviewers 2 and 3 that correction for multiple tests needs to be addressed. In the revised results section, we now present results corrected for multiple tests. See new table 2 and page 3 and 4:

“Table 2 depicts Pearsons’ correlations of tactile performance with personality dimensions. Results demonstrate positive correlations of tactile acuity with empathy (empathic concern), but not with SPS or other personality dimensions (Bonferroni corrected, adjusted p-value of 0.005).”

“Left and right tactile acuity correlated with r = 0.68 (similar to our previous study (Schaefer, Joch, & Rother, 2021)).”

Furthermore, we now changed table 3 to report the correlations for left and right hand separately (instead of the mean of both hands in the previous version).

Computing separate analyses for left and right hand was motivated by our previous study, which found different results for the association of left and right hand with empathy (Schaefer et al., 2021). In the revised version we now address this point in the methods as well as in the discussion section. We added on page 3:

“Furthermore, we computed separate analyses for left and right hand because our previous study found different results for left and right body side (Schaefer, Joch, et al., 2021).”

We added in the discussion section (page 9):

“Similar to our previous study we found relationships of tactile thresholds with empathy only for the right hand (Schaefer, Joch, et al., 2021). What is the reason for this laterality? Although it is well-known that the right side of our brain is linked to social perceptions, this does not mean that the left hemisphere is not important for processing social information. For example, it has been suggested that the left side of our brain is linked to emotions related to engaging and approaching, e.g., happiness when viewing a smile, whereas the right hemisphere may be associated with emotions related to avoidance, for example, fear (Brookshire & Casasanto, 2018). Other reasons may point to handedness (all of our participants were right-handed). Future studies are needed to address this finding. “

  1. “My other major concern is that the extant research is not discussed in a clear way. The paper includes SEP work from 1965, but EEG has evolved quite a bit since then. They should include similar studies from the last 10 years and build their arguments based on these..”

 We agree with the reviewer and added more recent paper in our discussion section. See pages 8-9:

“In addition, it has been shown that dispositional empathy predicted activity in primary somatosensory cortex when receiving touch by a hand (Schaefer, Kühnel, Rumpel, & Gärtner, 2021). Furthermore, structural brain differences in somatosensory brain areas have been linked to dispositional empathy (Banissy, Kanai, Walsh, & Rees, 2012). Moreover, several recent studies using imaging methods demonstrate a relationship between empathy and social touch (Goldstein, Shamay-Tsoory, Yellinek, & Weissman-Fogel, 2016; Peled-Avron, Goldstein, Yellinek, Weissman-Fogel, & Shamay-Tsoory, 2018; Peled-Avron, Levy-Gigi, Richter-Levin, Korem, & Shamay-Tsoory, 2016; Shamay-Tsoory & Eisenberger, 2021), which is also supported by qualitative approaches, for example, in physicians (e.g., (Kelly, Svrcek, King, Scherpbier, & Dornan, 2020). Thus, empathy and touch seem to be closely associated.”

“Furthermore, our results are also in line with more recent studies suggesting relationships between early somatosensory processing and personality or dispositions to develop diseases (Hagenmuller et al., 2019; Norra et al., 2004; Pavony & Lenzenweger, 2013, 2014; Rigato, Bremner, Gillmeister, & Banissy, 2019).”

Minor comments:

P1, L17 Remove (IRI)

 P1, L17 Remove (HSPS)

P1, L27  “to emerge” – remove -d

P1, L17 “that developed a type of tactile sense”, the tactile sense consists of several receptor types e.g. Meissner, Ruffini, Pacinian etc.

P1, L35 interact instead of act

P2, L48 “hypothesized”

P2, L50 The somatosensory system is more than just the cortex, please rephrase

P2, L51 in the postcentral gyrus (not on)

P2, L56 we are god at (not in)

All done or corrected.

P2, L57 You list studies (11-13) which seem very relevant. These should be described in detail

We added: “Moreover, it has been shown that dispositional empathy is associated with performance in a tactile acuity task, thereby suggesting that we may emotionally understand the other better when we are good at processing tactile stimuli felt on our own body (Schaefer, Joch, et al., 2021). Furthermore, research in mirror touch synesthesia similarly shows that in some individuals an observed tactile sensation can produce a felt tactile sensation on their own body, suggesting that we empathize with others through a process of simulation (Banissy, Walsh, & Ward, 2009; Banissy & Ward, 2007).”

P2, L63 Since you decided to call the paper Orchids and Dandelions and include it here, maybe you should explain what it is

We added: “Lionetti et al. used these flower metaphors to characterize highly sensitive individuals as orchids and less sensitive people as dandelions (Lionetti et al., 2018).”

P2, L72

P2, L54 Taken together, more research is needed…

P2, L55 “or SPS to processing” (not with)

P3, L110 Spell out acronyms SPF and IRI

All corrected.

P3, L110 In abstract you say you used IRI not SPF

Corrected.

P3, L116 Fantasy is german, Imagination is a better word in English, methinks

P3, L118 Remove “According to Davis”

P3, L122 spell out HSPS

Corrected.

P3, L126 Why not FS, PT, PD?

We did consider these dimensions in the subsequent analyses.

P4, Do you need correction for multiple comparison?

See above, point 1.

P4, L151 Why not combine left and right hands?

See above. point 1.

P6 Discuss the logic of logic at handedness separately and why right hand

Done, see above.

P6, Discuss how the many tests can influence results

We added on page 9: “Last, our study included many correlations, which increase the risk of type I error. We addressed this problem by adjusting probability values using Bonferroni correction.”

P7, 214 stained? did you mean sustained?

Corrected.

P7, 215 the body part in focus

Corrected.

P7, 240 some of this research is quite ancient (Shagass and Schwarz, 1965). SEPs looked quite differ back then. Our modern systems are much better than 1965. Include more modern work.

Done, see above, point 2.

Reviewer 3 Report

This is a nice study looking at the relationship of empathic abilities with tactile acuity. It suffers from the typical issues (small sample size, correlational design, multiple comparisons), which are addressed in the discussion. My main concern is the missing correction for multiple correlations. Also, the finding of a relationship of tactile acuity with empathic abilities but not with the sensitivity score is somewhat surprising. This should be discussed more. It seems intuitive that acuity should relate to the sensitivity score - so if it doesn't, maybe the problem is the sensitivity score (for example because of self report)? 

Comments:

Not sure about the title. What do you mean with orchids and dandelions? I had to look it up and found a reference to a book on raising children... I see that you mention it again in the introduction, but it would be very helpful if you would explain the meaning of this metaphor.

The abstract needs to be more concise. 

Intro:

"The tactile sense may have enabled simple invertebrates not only to receive information about the physical nature of the close environment but also to enable the first basic forms of social action." This is a weird sentence with two times "enable". 

With regard to the role of touch, you could refer to some literature on tactile communication, e.g. McIntyre, S., Moungou, A., Boehme, R., Isager, P. M., Lau, F., Israr, A., ... & Olausson, H. (2019, July). Affective touch communication in close adult relationships. In 2019 IEEE World Haptics Conference (WHC) (pp. 175-180). IEEE.

and

App, B., McIntosh, D. N., Reed, C. L., & Hertenstein, M. J. (2011). Nonverbal channel use in communication of emotion: how may depend on why. Emotion11(3), 603.

"extraverts" --> extroverts is the more common spelling.

The expressions "extrovert" and "introvert" are quite collegial. Maybe start by introducing the concepts extraversion and intraversion.

"thereby 55 suggesting that we may emotionally understand the other better when we are good in processing tactile stimuli felt on our own body" --> if we are good at

"in environmental sensitivity" what does this mean? sensitivity to external stimuli? I saw that this expression is used in the reference you cite, but I find the expression confusing since uncommon. Maybe rephrase or explain.

"personality traits such as empathy or SPS" Empathy isn't a personality trait, "empathic abilities" maybe. SPS, you said yourself, is a controversial concept, even more so to consider it a personality trait. Especially when taking into account that Aron&Aron built a whole business on this, their own, claim...

line 76 "fill"

Methods:

line95 "on the palm"

line 101 "consisted of"

l 105 "describes"

Results:

Considering that there is no difference in tactile acuity for left and right hand, you could combine them in the same analysis. It's surprising that the acuity is not different between left and right hand, but then your linear model is less significant for the left. This should be addressed in the discussion. 

In Table 2, is this tactile acuity for left or right or combined? Is this corrected for multiple correlations? 

"Big Five personality measures did not correlate with general tactile acuity, but with 161 tactile thresholds of the right hand. Here neuroticism and openness were positively linked 162 with tactile thresholds (neuroticism: r = 0.32, p = 0.04; openness: r = 0.35, p = 0.03)." also here: this will not survive correction for multiple comparisons. 

Discussion

"Thus, the current results extend previous findings by showing that tactile thresholds of other body parts also seem to be linked to empathic personality traits." Only partially, because you do not find this for the left hand. 

l 204: "on other’s bodies" --> others'

"Previous research found support for this view by showing that brain regions known to be related to awareness and empathy as well as higher-visual processing are increased in individuals with high SPS" Sentence needs to be revised.

General: only introduce abbreviations if you actually use them. 

Author Response

Comments of reviewer #3:

We would like to thank the reviewer for the positive and helpful comments on our manuscript.

  1. „ My main concern is the missing correction for multiple correlations..” 

 This is an important point. The revised version now reports corrected data (see also our response to reviewer 2, point 1). See now table 2 and pages 3 and 4:

“Table 2 depicts Pearsons’ correlations of tactile performance with personality dimensions. Results demonstrate positive correlations of tactile acuity with empathy (empathic concern), but not with SPS or other personality dimensions (Bonferroni corrected, adjusted p-value of 0.005).”

Furthermore, on page 6:

“Last, our study included many correlations, which increase the risk of type I error. We tried to address this problem by adjusting probability values using Bonferroni correction.”

  1. “Also, the finding of a relationship of tactile acuity with empathic abilities but not with the sensitivity score is somewhat surprising. This should be discussed more. It seems intuitive that acuity should relate to the sensitivity score - so if it doesn't, maybe the problem is the sensitivity score (for example because of self report)?“ 

 We agree with the reviewer that this point is important (see also reviewer 1). We address this lack of finding now more in detail in the discussion section, pointing to the possible heterogenous characteristic of the SPS questionnaire and also to the need of testing other questionnaires that may focus more directly on sensory perception. We added on page 8:

“Given that SPS seems to be a rather heterogeneous concept addressing several distinct dimensions, future studies should consider other and newer questionnaires to measure (hyper)sensitivity more directly, for example, the Glasgow Sensory Questionnaire (Robertson & Simmons, 2013) or the Sensory Perception Quotient (Tavassoli et al., 2014; Taylor et al., 2020; van Leeuwen et al., 2019). ”

  1. “Title: What do you mean with orchids and dandelions? I had to look it up and found a reference to a book on raising children... I see that you mention it again in the introduction, but it would be very helpful if you would explain the meaning of this metaphor.”

 This is a good point, we now explain this metaphor on page 2:

We added: “Lionetti et al. used these flower metaphors to characterize highly sensitive individuals as orchids and less sensitive people as dandelions (Lionetti et al., 2018).”

  1. “The abstract needs to be more concise.“

We followed the suggestion of the reviewer and made the abstract more concise and precise (see revised abstract).

  1. “"The tactile sense may have enabled simple invertebrates not only to receive information about the physical nature of the close environment but also to enable the first basic forms of social action." This is a weird sentence with two times "enable".

 We corrected this sentence to:

„Through the sense of touch, simple invertebrates not only received information about the physical condition of the immediate environment, but also the possibility of the first basic forms of social action. “

  1. “With regard to the role of touch, you could refer to some literature on tactile communication, e.g. McIntyre, S., Moungou, A., Boehme, R., Isager, P. M., Lau, F., Israr, A., ... & Olausson, H. (2019, July). Affective touch communication in close adult relationships. In 2019 IEEE World Haptics Conference (WHC) (pp. 175-180). IEEE. and App, B., McIntosh, D. N., Reed, C. L., & Hertenstein, M. J. (2011). Nonverbal channel use in communication of emotion: how may depend on why. Emotion, 11(3), 603.“

Thanks for drawing our attention to these papers. We added this work in the introduction.

  1. “"extraverts" --> extroverts is the more common spelling.“

Corrected.

  1. „The expressions "extrovert" and "introvert" are quite collegial. Maybe start by introducing the concepts extraversion and intraversion.“

We agree and corrected this paragraph.

  1. "thereby 55 suggesting that we may emotionally understand the other better when we are good in processing tactile stimuli felt on our own body" --> if we are good at“

Corrected.

  1. „"in environmental sensitivity" what does this mean? sensitivity to external stimuli? I saw that this expression is used in the reference you cite, but I find the expression confusing since uncommon. Maybe rephrase or explain.“

We agree and reframed the term to “external”.

  1. “"personality traits such as empathy or SPS" Empathy isn't a personality trait, "empathic abilities" maybe. SPS, you said yourself, is a controversial concept, even more so to consider it a personality trait. Especially when taking into account that Aron&Aron built a whole business on this, their own, claim...“

These are good points. We are now more concrete here and added:

“Furthermore, the proposed concept of SPS is still controversial discussed, in particular with respect to consider it as a personality trait”

“Taken together, more research is needed to determine the relationship of empathic abilities and SPS to processing in the tactile modality.”

  1. “line 76 "fill", line95 "on the palm". line 101 "consisted of",105 "describes"“

Corrected.

  1. “Considering that there is no difference in tactile acuity for left and right hand, you could combine them in the same analysis. It's surprising that the acuity is not different between left and right hand, but then your linear model is less significant for the left. This should be addressed in the discussion.“

This is an important point (see also our response to reviewer 2, point 1).  We added in the discussion section, page 9:

“Similar to our previous study we found relationships of tactile thresholds with empathy only for the right hand (Schaefer, Joch, et al., 2021). What is the reason for this laterality? Although it is well-known that the right side of our brain is linked to social perceptions, this does not mean that the left hemisphere is not important for processing social information. For example, it has been suggested that the left side of our brain is linked to emotions related to engaging and approaching, e.g., happiness when viewing a smile, whereas the right hemisphere may be associated with emotions related to avoidance, for example, fear (Brookshire & Casasanto, 2018). Other reasons may point to handedness (all of our participants were right-handed). Future studies are needed to address this finding.“

In addition, in the methods section (page 3) we added: 

“Furthermore, we computed separate analyses for left and right hand because our previous study found different results for left and right body side (Schaefer, Joch, et al., 2021).”

  1. “In Table 2, is this tactile acuity for left or right or combined? Is this corrected for multiple correlations?“

We now refer to left and right acuity separately in table 2. The revised table now presents corrected values.

  1. „"Big Five personality measures did not correlate with general tactile acuity, but with 161 tactile thresholds of the right hand. Here neuroticism and openness were positively linked 162 with tactile thresholds (neuroticism: r = 0.32, p = 0.04; openness: r = 0.35, p = 0.03)." also here: this will not survive correction for multiple comparisons.“

We agree with the reviewer and changed this parapgraph to:

„Big Five personality showed positive relationships with tactile thresholds of the right hand (neuroticism: r = 0.32, openness: r = 0.35), but failed to reach the level of significance.”

  1. „"Thus, the current results extend previous findings by showing that tactile thresholds of other body parts also seem to be linked to empathic personality traits." Only partially, because you do not find this for the left hand“

This is correct, we changed the statement to:

“Thus, the current results extend previous findings by showing that tactile thresholds of other body parts also seem to be linked to empathic personality traits (at the right side of the body).”

  1. “l 204: "on other’s bodies" --> others'“

Corrected.

  1. “"Previous research found support for this view by showing that brain regions known to be related to awareness and empathy as well as higher-visual processing are increased in individuals with high SPS" Sentence needs to be revised.“

We changed the statement to:

“Previous research found support for this view with respect to visual processing and awareness”

  1. “General: only introduce abbreviations if you actually use them. „

We revised the manuscript with to respect to this point.